# Reported Injuries from Sharp Objects among Healthcare Workers in Central Greece

**DOI:** 10.3390/healthcare10071249

**Published:** 2022-07-04

**Authors:** Anna Patsopoulou, Ioannis Anyfantis, Ioanna V. Papathanasiou, Evangelos C. Fradelos, Maria Malliarou, Konstantinos Tsaras, Foteini Malli, Dimitrios Papagiannis

**Affiliations:** 1Faculty of Nursing, School of Health Sciences, University of Thessaly, 41500 Larissa, Greece; efradelos@uth.gr; 2European Agency for Safety and Health at Work (EU-OSHA), 48003 Bilbao, Spain; anyfantis@osha.europa.eu; 3Community Nursing Laboratory, Faculty of Nursing, University of Thessaly, 41500 Larissa, Greece; iopapathanasiou@uth.gr; 4Laboratory of Education, Research of Trauma Care and Patient Safety, Faculty of Nursing, University of Thessaly, 41500 Larissa, Greece; mmalliarou@gmail.com; 5Public Health & Vaccines Laboratory, Faculty of Nursing, School of Health Sciences, University of Thessaly, 41500 Larissa, Greece; ktsa@uth.gr (K.T.); dpapajon@gmail.com (D.P.); 6Respiratory Disorders Lab, Faculty of Nursing, University of Thessaly, 41500 Larissa, Greece; mallifoteini@yahoo.gr

**Keywords:** needle stick, sharp injuries, needle recapping, healthcare workers

## Abstract

Sharp injuries (SIs) are incidents or accidents caused by a needle, blades (such as scalpels) or other medical instruments which penetrate the skin. They are among the major work-related injuries in healthcare professionals. The purpose of this study is to estimate SIs in healthcare workers (HCWs) in Central Greece. Method: A cross-sectional descriptive study through an online survey in healthcare facilities in Central Greece was conducted. Snowball sampling contributed to further dissemination of the survey among the target population. The modified version of the EPINet questionnaire was used with self-reported answers of the participants via electronic Google form. Results: Analysis of collected data indicated that 74.1% of the participants had at least one injury, with the highest number of injuries occurring in nursing staff at 65.1% and 62.3% of injuries recorded in the morning shift. With respect to the site of the injury, participants reported 33.1% of the injuries in the patient’s room, 11.8% in the nurse’s station, 9.6% in the Emergency Department (ED), 9.2% in the Intensive Care Unit (ICU), 8.4% in blood sampling, 8.4% in surgery, and only 7.8% in laboratories or other places. Additionally, hands were the most frequently affected body part (96%), while 69.6% of the workers did not report the injury and 53% of them did not apply the procedures and guidelines defined by the healthcare organization (employer). Relative factors to the injury are age, level of education, shifts, and possibly sex. Conclusions: SIs are the “Achilles heel” of health workers. The high incidence and low reporting rate of SIs highlights the need for specialized training and education. Age, work experience, and shift appear to significantly affect the incidence of injury.

## 1. Introduction

The health and safety of health workers is imperative both for the quality of working conditions and for the quality of patient care. Healthcare workers, according to Martins and Co (2012), belong to those at the highest risk of occupational infection with biological agents, as they are exposed to bodily fluids daily [1]. An important component for health workers is that daily contact with patients results in a small mistake or carelessness being potentially fatal to their safety every shift [2].

There are many factors that endanger the health and safety of workers in the workplace of healthcare. One of the primary factors is an injury from a sharp object. According to the Center for Disease Control (CDC), a sharp injury is an incidence caused by medical instrument which penetrates the skin [3]. A SI refers to wound from a needle, scalpel, or other sharp object that may result in exposure to blood or other bodily fluids [3]. 

Every year, thousands of health workers are exposed to dangerous and deadly pathogens transmitted to the blood through infected sharp objects during daily clinical procedures [4]. These reports of bloodborne infections may carry the risk of infection with hepatitis B, hepatitis C, and human immunodeficiency virus [5,6]. In Europe, 1,000,000 annual SI cases are estimated among healthcare workers [7]. These injuries from sharp objects have been reported as the most common occupational hazards faced by workers in healthcare facilities, including doctors, nurses, paramedics, cleaning workers, etc. [8]. Several studies claim that among health professionals (nursing staff, medical staff, laboratory workers, hospital cleaning workers, and students, who are usually trained in health settings) there is a differentiation in reporting of needle injuries or sharp objects, with nursing staff demonstrating the highest rates [9,10,11], both internationally and for the case of Greece [12]. This could likely be justified by the fact that nursing staff are the professional category devoting the most time to patient care, and they experience high rates of exposure to infection or injury [2,13]. The likelihood of injuries with sharp objects is high with more frequent needle-related injuries, estimated at 400,000 a year in the US, and it has been discovered that about one third of nurses are injured with a needle at least once per year [14]. Of these incidents, at least half are unreported and are not recorded; the unreported rate is between 26 and 85% [15,16]. Previous research has identified that many workers do not report injuries because they underestimate the risk of infection, consider the reporting process to be time-consuming, or due to the fear of being considered unprofessional [17].

The primary objective of this study was to estimate the prevalence of SIs among healthcare workers (HCWs) in central Greece. Furthermore, our study investigated the SI risk factors, the exposure conditions, the nature of work, the demographic characteristics, and the work experience.

## 2. Methods 

### 2.1. Study Design

The survey included the five public hospitals of Central Greece (University General Hospital of Larissa, General Hospital of Larissa, General Hospital of Volos, General Hospital of Karditsa, and General Hospital of Trikala). The sample of the study consisted of people working in healthcare facilities. The survey was conducted between January and February 2019. A total of 2400 HCWs (data from each hospital used to submit the online form of the questionnaire), only 457 were participated in the survey. The survey was completed by 345 women and 107 men, and we lost 5 because of incomplete information. This resulted in a response rate of 19%. 

### 2.2. Questionnaire 

A modified version of the Exposure Prevention Information Network (EPINet) Report for Needle stick and Sharp Object Injuries Questionnaire [18] was translated into Greek. It includes 24 questions aimed at investigating sharp injuries to people working in healthcare facilities. The answers provided are self-quoted values of the respondents. The questionnaire was submitted online and via Google form. The questionnaire was anonymous and voluntary. The internal consistency of the modified version of the questionnaire EPINet was assessed using the Cronbach’s alpha factor, which was discovered to be a = 0.777.

### 2.3. Ethical Considerations

Data collection and analysis were designed to ensure data confidentiality and were conducted in accordance with national and European laws (GDPR, 2018) and the Personal Data Act (523/1999). The study was approved by the Ethics Committee of the Technological Educational Institute of Thessaly (Protocol Number: 2344/14-06-2018). The electronic data were saved in a protected folder, accessible only by the principal investigator. On the first screen of the survey, participants were shown a statement that included details of the study and data handling. Participants were asked to provide written informed consent for participation.

### 2.4. Data Analysis

Qualitative variables (demographic characteristics, type of injuries, etc.) were described as frequencies with percentages. Associations between binary and quantitative variables were conducted using independent-sample *t*-tests. The Kolmogorov–Smirnov test was used to check the normality of the quantitative data.

Differences and relationships between categorical factors were tested using univariable analysis, calculating the relative risks (RR) and the corresponding 95% confidence intervals (95% CIs). 

The level of statistical significance was set at a *p* value of 0.05. Data were analyzed using SPSS 19.0 (IBM SPSS Inc., Armonk, NY, USA: IBM Corp, USA).

## 3. Results

### 3.1. Sociodemographic Characteristics

A total of 457 eligible health workers were included in the study. Of these, the majority (452 health workers) voluntarily agreed to participate in this study, and 5 submitted largely incomplete questionnaires. Moreover, 345 participants were female and 107 were male. The age of the participants included in this study ranged between 20 and 55+ years old with a mean age of 40 (SD = ±6.122) (Table 1).

### 3.2. Sharp Injury (SI)

Among the respondents, 335 (74.1%) encountered SIs in the past year. Of these, 49.3% were due to needles, 14.3% were due to more than one sharp object, 11.9% were due to other medical devices, 10.7% were due to syringes, 4.2% were due to surgical sutures, 2.4% were due to blood glucose lancets and catheters, and 5.6% were due to medical equipment (blades, insulin pen, ampule, glass). About 48.6% of SIs occurred when using a single pair of gloves. Additionally, 78.5% of the sources patients were identifiable, and 70.2% of the participants caused the injury on their own. Finally, another important finding was the high percentage (69.6%) of unreported injuries by the HCW (Table 2).

### 3.3. Work Environment Related Factors for SIs

Based on collected data, 33.1% of SIs happen in the patient room and 33.8% of respondents reported that the injury occurred after use, while disassembling a device or equipment. Additionally, 62.3% of the HCW were injured during the morning shift and 22.1% were injured during the night shift (Table 3). 

### 3.4. Factors Associated with Needle Stick and Sharp Injuries

In bivariate logistic regression analysis, sex of the respondent, age, work experience, shift, and profession were found to be statistically significantly associated with SIs (*p* < 0.05) (Table 4). After bivariate analysis, only those variables which were significantly related (𝑝 value < 0.05) were entered for further multivariate analysis. By adjusting potential confounders in multivariate logistic regression analysis, only years of work, age, shift, and profession were significantly associated with SIs. The participants’ sex was not significantly associated with SIs in multivariate analysis. Healthcare providers with less experience (years of work) were 99% more likely to face SIs [OR = 1.38 (1.32,1.44)] as compared to those who had more than 10 working years. Moreover, health workers aged over 26 years old were less likely to encounter SIs than younger participants (18–25 years old) [OR = 1.62 (1.52,1.71)]. Night shift workers (23–07) were more likely to encounter SIs [OR = 1.10 (0.99,1.22)] than other shifts. Lastly, health workers (doctors, nurses, and midwives) had a better knowledge on how to safely handle a sharp object, compared to those who did not have related knowledge (Laboratories/Technician, students, and other professions). 

## 4. Discussion

SIs are frequent preventable occupational hazards among HCW. According to World Health Organization estimates, approximately two million SI cases are reported annually. This number could be an underestimation as many cases of SIs are not reported, particularly in low-income countries [19,20,21,22,23,24,25]. 

The present study indicated high prevalence of SI. In this study, 74.1% of the respondents had an SI at least once in the previous one year. This finding is like those reported in South Korea (70.4%) [26], Ethiopia (60.2%) [27], and Nigeria 55.8% [19]. Our discovered prevalence is higher than a study in Saudi Arabia which estimated 22.4% of healthcare workers had at least one event of SI [25]; Iran had 42.5% [28], Germany 28.7% [29], Turkey 20.7% [30], and China had 27.5% of worker’s experience SI cases [31]. A very low rate of 8.4% was reported in Damman [11]. 

Our findings also indicated that the highest rate of SIs (65.1%) was in nursing staff, which is much higher than other studies internationally. In an analysis on retrospective data from 732 and prospective data from 960 nurses on needle stick exposures, nurses from units with low staffing and poor organizational climates were generally twice as likely as nurses on well-staffed and better-organized units to report risk factors, needle stick injuries, and near misses [9]. A study conducted in Kampala, Uganda presented a high percentage (57%) of nurses and midwives who had experienced at least one needle stick injury in the last year [10]. Similar results for the profession of nurses was presented by the Alfulayw et al. in Saudi Arabia, where the most affected staff was nurses (52.5%), injuries commonly resulting from disposing of syringes (58.9%). In contrast, the incidence of NSIs among doctors was 24.9% [11]. Other studies in Saudi Arabia by Albeladi et al. (2021) reported 38.4% of nurses [32] injured from SIs, and Abalkhail et al. (2022) reported 34.8% of nurses injured [25]. An explanation of this could be directly dealing with sharp objects at work, which is significantly associated with the SI experience. Age is also a significant predictor of the risk of SIs; it was found that participants between 18–25 years old were significantly associated with higher risk of SIs compared to other aged groups. Our survey also discovered a significant relationship between years of work experience and shift, but a lack of significant association between gender. It was discovered that 62.3% of the SI incidents took place in the morning shift. A possible explanation of this could be that most medical procedures take place in the morning, so more HCWs are needed in the morning shift.

With respect to the healthcare situation in our country, Greece, an older study conducted by Pournaras et al. indicated that the overall injury rate of participants was 2.4% per year and, of the total incidents, 52.8% were reported by nurses, 27.1% by MDs, followed by housekeeping workers with 14.4% [12]. These findings are aligned with our study. Nevertheless, the annual SIs cases were extremely high, a fact that must be considered for a future national survey for SIs. 

Analogous to other countries, the prevalence of SIs among healthcare workers in Central Greece was relatively high [20-25]. The most important factors that cause SIs were due to needles, other medical devices, syringes, surgical sutures, blood glucose lancets, catheters, and medical equipment (blades, insulin pens, ampules, glass). About 46.2% of SIs occurred when using a single pair of gloves. According to a survey administered in the UK to a sample of nursing students in a university, sharp injuries were most likely to occur with glass ampoules when preparing injections in the second year of the program. Contributing factors to sharps injury were identified, with inexperience being the primary cause [33]. Moreover, at least half of SIs are not reported or recorded, while the non-reporting rate according to previous research was between 26 and 85% [15,16]. A significant problem for occupational medicine is the underreporting of occupational injuries and illness in health facilities. In our study, only 30.4% mentioned the injury because they underestimated the risk of infection or considered themselves at risk of being characterized as unprofessional [17]. Older studies specific to the healthcare facilities found that 40% of hospital service workers had not reported one or more injuries even though two of three of these unreported injures required medical care and half of them resulted in lost work time [34]. 

Previous studies indicate that HCW are aware of the benefits of early reporting, although a culture of silence persists [35]. In the present study, we recorded a high percentage (69.6%) of unreported injuries in healthcare workers. Similar findings of unreported injuries were reported by Abalkail et al. at 53.8% of the cases, [25] in addition to Elmiyeh et al. with a study conducted in the UK, where only half of those who had been affected had reported all injuries [36]. This lack of reporting could also be due the fact that HCWs do not perceive the injury from a sharp object as severe. The principal reason for non-reporting was a low perceived risk of transmission of infection. Almost the vast majority in the study acknowledged the benefits of early reporting concerning themselves, but only 61% thought that early reporting would benefit the patient [36]. A recent study from Saudi Arabia revealed that nearly two-thirds (68.5%) of the study population were exposed to one or more biological health hazards including needle stick injuries. These biological health hazards were significantly higher among the HCWs working in higher centers, tertiary, or specialty hospitals [37].

## 5. Conclusions

In our study, we reported a high incidence of SIs (74.1%) and a low reporting rate (30.4%). The most affected group was nurses, followed by doctors. In Greece, SI cases were extremely high, a fact that must be considered for a future national survey on SIs. SIs should be considered in the greater context of safety and health management, in addition to preventive strategies. All the above could contribute to the reduction of the high SI rates reported and improvement of occupational safety and health for this category of workers. The two different aspects of proactive approach (to reduce the rate of SIs) and reactive approach (to reduce the consequences of SIs through early response) should be discussed and highlighted as well. Infections caused by occupational exposures are costly in terms of human suffering, the socio-economic impact, and the financial responsibilities borne by accident insurance institutions. It is important that healthcare providers receive training to fill the skill gap and identify the trends of SIs.

## 6. Limitations

Our study has several limitations. The study was questionnaire-based and there was a potential for information bias to occur. The possibility of providing invalid answers from the participants is not excluded. Furthermore, respondents may not have answered truthfully, particularly on sensitive professional questions. The low response rate of the present survey is another limitation, and the sample may not be representative of all HCWs. Another major limitation is that no qualitative data were obtained on the reasons for non-reporting.

## Figures and Tables

**Table 1 healthcare-10-01249-t001:** Sociodemographic characteristics of healthcare workers in Central Greece.

Variable	Frequency (%)
Sex	
Male	107 (23.7%)
Female	345 (76.3%)
Age	
20–25	105 (23.2%)
26–35	95 (21.0%)
36–45	160 (35.4%)
46–55	76 (16.4%)
+55	16 (3.5%)
Educational Level	
College	252 (55.8%)
Master	103 (22.8%)
PhD	19 (4.2%)
High School	52 (11.5%)
Other	26 (5.8%)
Profession	
Physician	57 (12.6%)
Nursing	193 (42.7%)
Students	59 (13.1%)
Nursing Assistant	53 (11.7%)
Midwifery	10 (2.2%)
Laboratory	50 (11.1%)
Other	30 (6.6%)
Work Experience	
0–10	252 (55.8%)
11–20	131 (29.0%)
21–30	53 (11.7%)
>30	16 (3.5%)
Work Department	
Clinic	141 (31.2%)
ICU	62 (13.7%)
ED	39 (8.6%)
Laboratories	70 (15.5%)
Surgery	47 (10.4%)
Other	93 (20.6%)

**Table 2 healthcare-10-01249-t002:** SIs are among healthcare professionals.

Variable	Frequency (%)
Did you encounter SIs;	
Yes	335 (74.1%)
No	117 (25.9%)
Was source patient identifiable;	
Yes	263 (78.5%)
No	72 (21.5%)
Did you cause the injury;	
Yes	234 (70.2%)
No	101 (29.8%)
The item was	
Contaminated	120 (35.8%)
Uncontaminated	152 (45.4%)
Unknown	63 (18.8 %)
Which device caused the injury	
Surgical equipment	2 (0.6%)
Glass	2 (0.6%)
Medication ampule	4 (1.2%)
Insulin Pen	4 (1.2%)
Blades	4 (1.2%)
Blood Glucose Lancet	8 (2.4%)
Catheters	8 (2.4%)
Surgical Suture	14 (4.2%)
Syringe	36 (10.7%)
Other	40 (11.9%)
>more than one answer	48 (14.3%)
Needles	165 (49.3%)
Location of the injury	
Right Hand	137 (40.9%)
Left Hand	96 (28.7%)
Both Hands	90 (26.8%)
Other	12 (3.6%)
If injury was to a hand, did the sharp item penetrate?	
Single pair of gloves	163 (48.6%)
Double pair of gloves	24 (7.2%)
No gloves	56 (16.7%)
Sometimes I wear gloves, sometimes no	92 (27.5%)
Did you report the injury?	
Yes	102 (30.4%)
No	233 (69.6%)

**Table 3 healthcare-10-01249-t003:** Working environment conditions.

Variable	Frequency (%)
Work Shift	
Morning	209 (62.3%)
Evening	74 (22.1%)
Night	29 (8.7%)
All	23 (6.9%)
Where did the injury occur;	
Patient room	111 (33.1%)
Nurses station	40 (11.8%)
Emergency Department	32 (9.6%)
ICU	31 (9.2%)
Blood Sampling	28 (8.4%)
Surgery	28 (8.4%)
Other	26 (7.8%)
Xray room	14 (4.2%)
Blood Bank	12 (3.6%)
Most of the places	9 (2.7%)
Recovery room	4 (1.2%)
The purpose was the SIs used	
Artery line	5 (1.5%)
Central Venus line/IV catheter	11 (3.3%)
Unknown	13 (3.9%)
IM (Intra-muscular) inject	14 (4.2%)
>more than one answer	22 (6.6%)
Sc (Subcutaneous) injects	33 (9.9%)
Other	37 (11.0%)
Surgical Suture	41 (12.2%)
Blood Glucose	43 (12.8%)
Peripheral Venus line	49 (14.6%)
Blood sampling	67 (20.0%)
When did the injury occur;	
While withdrawing SIs	2 (0.6%)
After use, while recapping	16 (4.8%)
>more than one answers	44 (13.1%)
During use of item	47 (14.1%)
Other	53 (15.8%)
Before use	55 (16.4%)
After use, while disassembling device or equipment	118 (35.2%)

**Table 4 healthcare-10-01249-t004:** Bivariate and multivariate logistic regression analysis of factors associated with SIs.

Sharp Injuries (SIs)					
Variable	Response	Yes	No	OR (95% CI) *	*p* Value
**Sex**	Male	84 (25.1%)	23 (19.7%)	1.36 (0.8,2.3)	0.234
	Female	251(74.9%)	94 (80.3%)	1	
**Work experience**	0–10	157 (46.9%)	95 (81.1%)	1.38 (1.32,1.44)	**0.004**
	11–20	117 (34.9%)	14 (12.0%)	1.11 (1.05,1.16)	0.690
	21–30	46 (13.7%)	7 (6.0%)	1.1 (1.04,1.23)	0.561
	>30	15 (4.5%)	1(0.9%)	1	
**Age (years)**	18–25	40 (11.9%)	65 (55.6%)	1.62 (1.52,1.71)	**0.001**
	26–35	76 (22.7%)	19 (16.2%)	1.20 (1.12,1.28)	0.479
	36–45	139 (41.5%)	21 (17.9%)	1.13 (1.08,1.18)	0.952
	44–55	66 (19.7%)	10 (8.5%)	1.13 (1.05,1.21)	0.951
	>55	14 (4.2%)	2 (1.7%)	1	
**Shift** **(Hours)**	07–15	202 (62.5%)	7 (58.3%)	1.03 (1.01,1.06)	0.375
	15–23	72 (22.3%)	2 (16.7%)	1.03 (0.99,1.06)	0.515
	23–07	26 (8.1%)	3 (25.0%)	1.10 (0.99,1.22)	**0.037**
	All	23 (7.1%)	0 (0%)	1	
**Profession**	Physicians	46 (13.7%)	11 (9.4%)	1.16 (1.06,1.26)	0.838
	Laboratories/Technician	35 (10.4%)	15 (12.8%)	1.30 (1.17,1.33)	**0.015**
	Students	15 (4.5%)	44 (37.6%)	1.75 (1.64,1.86)	**0.001**
	Other	21 (6.3%)	9 (7.7%)	1.30 (1.13,1.27)	**0.051**
	Nursing/Nursing Assistant/Midwife	218 (65.1%)	38 (33.5%)	1	

* OR: odds ratio; CI: confidence interval.

## Data Availability

The data that support the findings of this study are available on request from the corresponding author.

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
