# Peer review of "Reported Injuries from Sharp Objects among Healthcare Workers in Central Greece"

_healthcare, 2022, doi:10.3390/healthcare10071249_

Round 1

Reviewer 1 Report

This research entitled “Reported injuries from sharp objects among health care workers in central Greece” written by Patsopoulou and his/her colleagues aimed to evaluate the prevalence of sharp injuries among HCP in central Greece. Considering the high requirements of this journal, this research needs to be improved and well revised before acceptance. Please check the following comments:

·      English language should be improved.

·      Please justify why most of the injuries recorded in the morning shift?.

·      Please write the full name of HCP in line 69.

·      Please name the 5 hospitals in the study design section.

·      In Table 2, 335 encountered SIs among the respondents but for other questions in Table 2 the total responses sometimes more than 335? And some questions the total responses less than 335? Please check again.

·      Please write the full name of NSI in line 248.

·      Please compare your findings with the recent published paper https://doi.org/10.3390/ijerph19106342

If you will answer to my requests, therefore, I think this manuscript could be accepted in this journal.

Author Response

Response to Reviewer 1 Comments

 Point 1: English language should be improved.

We used a Native English proofreader to improve the English syntax in our study.

Point 2:  Please justify why most of the injuries recorded in the morning shift?.

A possible explanation of this can be that most of the medical procedures take place in the morning shift, so more HCW needed in the morning shift.

Point 3: Please write the full name of HCP in line 69.

Done

Point 4: Please name the 5 hospitals in the study design section.

The survey included the 5 public hospital of Central Greece (University General Hospital of Larissa, General Hospital of Larissa, General Hospital of Volos, General Hospital of Karditsa, and General Hospital of Trikala).

Point 5: In Table 2, 335 encountered SIs among the respondents but for other questions in Table 2 the total responses sometimes more than 335? And some questions the total responses less than 335? Please check again.

We check all the tables as you can see in the revised paper

Point 6: Please write the full name of NSI in line 248.

Done

Point 7: Please compare your findings with the recent published paper https://doi.org/10.3390/ijerph19106342

Done

Reviewer 2 Report

The objective of this manuscript was to investigate Sharp injuries in health care workers in Central Greece, it is a well-written manuscript, with an adequate description of the methodology and that fulfills its objective, however, the information presented lacks originality, as the authors describe there is a large bibliography about injuries in health workers and the results presented could be of more than local interest.

It is a merely descriptive study, in which no scientific contribution is observed,  some observation is the following

1.The authors even mention a previous study carried out in the same geographical area, so I suggest that the differences found with respect to this one be discussed more deeply given the difference in years of perform.

Author Response

Response to Reviewer 2 Comments

 Point 1: The authors even mention a previous study carried out in the same geographical area, so I suggest that the differences found with respect to this one be discussed more deeply given the difference in years of perform.

Respone:

Regarding the healthcare situation in our country, Greece, an older study was conducted by Pournaras et al (only to one public hospital) showed that the overall injury rate of participants was 2.4% per year and of the total incidents (52.8%) were reported by nurses, (27.1%) by MDs, followed by housekeeping workers with (14.4%) percentage. In our study, we reported a high incidence of SIs (74.1%) and a low reporting rate (30.4%) (from 5 public hospitals). Nevertheless, the annually SIs cases were extremely high, a fact that must be considered for a future national survey for SIs.

Reviewer 3 Report

Thank you for the opportunity to review this study on sharp injuries in Central Greece. CHECK APPENDIX for all questions asked

Abstract:

  • Aim in abstract does not match aim at end of introduction; align. Refrain from using the word ‘evaluate’ in the aim as a questionnaire of 3 min duration is not an evaluation, but rather an analysis of some data. Furthermore, where are data shown that are an analysis of the established protocol? This has to be introduced as not known to the reader.
  • Conclusion is not based on the data shown in the manuscript regarding ‘reduced burnout may contribute to reducing injuries.’ Either add the data to prove this or remove it. Make conclusion specific to central Greece.

Introduction:

  • Line 47, state here the CDC definition for sharp injury.
  • For references used throughout the introduction, specify the context, e.g. line 53-57 reference 7, state needlestick injuries in European nurses working with patients who have diabetes from 2012 (as these data may have changed significantly in the ten year period since then).

Methods:

  • Established protocol not mentioned here, and is this a protocol for all of Greece?
  • State why hospitals in central Greece were included, why not all of Greece for example.
  • Explain if n=2400 was the sample of all HCP across the 5 hospitals combined.

Results:

  • Line 236, specify in the text if the 74.1% was an SI ‘within the past year’ and add this to the table, as per discussion line 280 this was within the past year
  • Table 2: spacing to be corrected for Yes/No on reporting
  • Line 268-271, where in the tables are the data to prove this statement that ’health workers had a better knowledge on how to safely handle a sharp object, compared to those who did not have related knowledge (Laboratories/ Technician, students and other professions)

Discussion:

  • First paragraph is background information that is more fitting for the introduction. Furthermore, avoid using the word ‘developing’ countries and replace with low-income or resource-limited.
  • Comparison of the data with studies from the African continent are not useful since the environment in which the health workers function is very different; comparison of data should be limited to within Europe or to other high income countries like Saudi Arabia as mentioned by the authors, while acknowledging that some of these studies are outdated
  • lines 299-301: Clarify why line 299 states ‘In our study…’ but a reference 16 to another study is placed at the end of the sentence line 301; suggest removing the reference as this reference is correctly placed in the introduction.
  • Lines 325-327 requires a reference to provide evidence for this statement.
  • No conclusion found, would be helpful for reader.

Limitations: add that a major limitation is that no qualitative data were obtained on the reasons for non-reporting and clarify why this was not done with focus group discussions or face to face interviews with health workers to inform prevention strategies.

Overall: moderate problems with grammar and syntax throughout. Native English proofreader required.

Author Response

Response to Reviewer 3 Comments

 Point 1:  Abstract:

  • Aim in abstract does not match aim at end of introduction; align. Refrain from using the word ‘evaluate’ in the aim as a questionnaire of 3 min duration is not an evaluation, but rather an analysis of some data. Furthermore, where are data shown that are an analysis of the established protocol? This has to be introduced as not known to the reader.
  • Conclusion is not based on the data shown in the manuscript regarding ‘reduced burnout may contribute to reducing injuries.’ Either add the data to prove this or remove it. Make conclusion specific to central Greece.

Response:

We make all the changes as you can see in the revised paper.

Point 2: Introduction:

  • Line 47, state here the CDC definition for sharp injury.

According to Center Disease Control (CDC) a sharp injury is an incidence, caused by medical instrument which penetrate the skin.

  • For references used throughout the introduction, specify the context, e.g. line 53-57 reference 7, state needlestick injuries in European nurses working with patients who have diabetes from 2012 (as these data may have changed significantly in the ten year period since then).

We change the reference.

Bouya, S.; Balouchi, A.; Rafiemanesh, H.; Amirshahi, M.; Dastres, M.; Moghadam, M.P.; Behnamfar, N.; Shyeback, M.; Badakhsh, M.; Allahyari, J.; et al. Global Prevalence and Device Related Causes of Needle Stick Injuries among Health Care Workers: A Systematic Review and Meta-Analysis. Ann. Glob. Health, 2020, 86, 35.

Point 3: Methods:

Established protocol not mentioned here, and is this a protocol for all of Greece?

Every hospital has its own protocol. We change it in our paper (we choose not to mention it).

  • State why hospitals in central Greece were included, why not all of Greece for example.

Because the 5 public hospitals in central Greece are a representative sample of greek public hospitals.

  • Explain if n=2400 was the sample of all HCP across the 5 hospitals combined.

Exactly. The permanent staff was 2400.

Point 4: Results:

  • Line 236, specify in the text if the 74.1% was an SI ‘within the past year’ and add this to the table, as per discussion line 280 this was within the past year

We change it

  • Table 2: spacing to be corrected for Yes/No on reporting

We change it

  • Line 268-271, where in the tables are the data to prove this statement that ’health workers had a better knowledge on how to safely handle a sharp object, compared to those who did not have related knowledge (Laboratories/ Technician, students and other professions)

In Table 4 (Bivariate and multivariate logistic regression analysis of factors associated with SIs).

The nurses and the doctors use sharp objects daily in addition to laboratories and students. Most of them had the skill and the knowledge how to handle with safe a sharp object.

Point 5: Discussion:

  • First paragraph is background information that is more fitting for the introduction. Furthermore, avoid using the word ‘developing’ countries and replace with low-income or resource-limited.

We change it

  • Comparison of the data with studies from the African continent are not useful since the environment in which the health workers function is very different; comparison of data should be limited to within Europe or to other high income countries like Saudi Arabia as mentioned by the authors, while acknowledging that some of these studies are outdated

We used more referneces from high income countries.

Wicker, S.; Jung, J.; Allwinn, R.; Gottschalk, R.; Rabenau, H.F. Prevalence and prevention of needlestick injuries among health care workers in a German university hospital. Int. Arch. Occup. Environ. Health, 2008, 81, 347–354.

Solmaz, M.; Solmaz, T. Experiences with Needle-stick and Sharp Object Injuries for Healthcare Workers in a State Hospital in Tokat Province, Turkey. Int. J. Occup. Hyg., 2017, 9, 142–148.

Cui, Z.; Zhu, J.; Zhang, X.; Wang, B.; Li, X. Sharp injuries: A cross-sectional study among health care workers in a provincial teaching hospital in China. Environ. Health Prev. Med. 201823, 2.

Alfulayw, K.H.; Al-Otaibi, S.T.; Alqahtani, H.A. Factors associated with needlestick injuries among healthcare workers: Implications for prevention. BMC Health Serv. Res. 202121, 1074.

Albeladi, O.; Almudaraa, S.; Alqusibri, A.; Alqerafi, N.; Alsenani, Y.; Saleh, E. Needle Stick Injuries among Health Care Workers in AL-Madinah AL-Munawara Governmental Hospitals in Saudi Arabia. Glob. J. Health Sci. 202113, 35.

Hambridge, K.; Endacott, R.; Nichols, A. Investigating the incidence and type of sharps injuries within the nursing student population in the UK. Br J Nurs., 2021, 30, 17, 998-1006.

Ashokkumar, T.; Khaloud Amash Hossin, A.; Ahmad Homoud, Al-H.; et al. Prevalence and Risk Factors of Occupational Health Hazards among Health Care Workers of Northern Saudi Arabia: A Multicenter Study. Int J Environ Res Public Health, 2021, 18, 21, 11489. doi: 10.3390/ijerph182111489

  • lines 299-301: Clarify why line 299 states ‘In our study…’ but a reference 16 to another study is placed at the end of the sentence line 301; suggest removing the reference as this reference is correctly placed in the introduction.

We change it

  • Lines 325-327 requires a reference to provide evidence for this statement.

We change it

  • No conclusion found, would be helpful for reader.

We add it

In our study, we reported a high incidence of SIs (74.1%) and a low reporting rate (30.4%). The most affected group was nurses followed by doctors. In Greece, SIs cases were extremely high, a fact that must be considered for a future national survey for SIs. Also, SIs should be considered as the greater context of safety and health management and preventive strategies. All the above could contribute to the reduction of the high SI rates reported and improvement of occupational safety and health for that category of workers. The two different aspects of proactive approach (to reduce the rate of SIs) and reactive approach (to reduce the consequences of SIs through early response) should be discussed highlighted as well. Infections caused by occupational exposures are costly in terms of human suffering, the social-economic impact, and the financial responsibilities borne by accident insurance institutions. It is important that health care providers should get training to fill the skill gap and being trained to identify the trends of SIs.

Point 6: Limitations: add that a major limitation is that no qualitative data were obtained on the reasons for non-reporting and clarify why this was not done with focus group discussions or face to face interviews with health workers to inform prevention strategies.

We add it. This is a fact that must be considered for a future national survey for SIs.

Point 7: Overall: moderate problems with grammar and syntax throughout. Native English proofreader required.

We used a Native English proofreader to improve the English syntax in our study.

Round 2

Reviewer 3 Report

No comments.